# Water Uptake as a Crucial Factor on the Properties of Cryogels of Gelatine Cross-Linked by Dextran Dialdehyde

**DOI:** 10.3390/gels7040159

**Published:** 2021-09-30

**Authors:** Natalia Volkova, Dmitriy Berillo

**Affiliations:** 1Department of Biotechnology, Lund University, P.O. Box 124, 22 100 Lund, Sweden; natalia.volkova@ftf.lth.se; 2Department of Solid State Physics, Lund University, P.O. Box 118, 22 100 Lund, Sweden; 3Department of Pharmaceutical and Toxicological Chemistry, Pharmacognosy and Botany School of Pharmacy, Asfendiyarov Kazakh National Medical University, Almaty 050000, Kazakhstan

**Keywords:** cryogels, water sorption, DVS, CLSM, BET, GAB, sorption parameters

## Abstract

We investigated the water sorption properties of macroporous cryogels of gelatine (Gel) and dextran dialdehyde (DDA) prepared via cryogelation at 260 K and following the freeze drying processes. Water vapour sorption isotherms for aerogels were studied at 293 K by two independent methods: static-gravimetric and dynamic vapour sorption (DVS) over a water activity range of 0.11–1.0. Experimental data were fitted by use of the Brunauer–Emmett–Teller (BET) and Guggenheim–Anderson–de Boer (GAB) models. The BET model (for a water activity range of 0.1 ≤ *p*/*p_o_* ≤ 0.5) was used to calculate the sorption parameters of the studied cryogels (the monolayer capacity, surface area and energy of interaction). In comparison with BET, the GAB model can be applied for the whole range of water activities (0.1 ≤ *p*/*p_o_* ≤ 0.95). This model gave an almost perfect correlation between the experimental and calculated sorption isotherms using nonlinear least squares fitting (NLSF). Confocal Laser Scanning Microscopy (CLSM) was used to confirm the structural differences between various DDA:Gel cryogel compositions. Thermogravimetric analysis and DSC data for aerogels DDA:Gel provided information regarding the bonded water loss, relative remaining water content of the material and the temperature of decomposition. Estimation of the amount of bound water in the cryogels after the freeze drying process as well as after the cycle of treatment of cryogels with high humidity and drying was performed using DSC. The results of the DSC determinations showed that cryogels with higher gelatin content had higher levels of bonded water.

## 1. Introduction

Cryogels are, in general, highly porous gel systems formed under special conditions, including two main steps: freezing and thawing. The structure and properties of obtained materials are affected by different factors: temperature, pH, solvent composition etc. The methods of production in various shapes and volume, properties and application areas of cryogels in biotechnology, biomedicine and bioremediation were recently described in several publications [1,2,3,4,5,6]. 

Different types of natural (chitine, cellulose, chitosan, gellan, alginate, dextran, collagen and gelatine) or synthetic polymers (polyethylenamine, PEGdimethacrylate, polyvinylalcohol etc.) can be used for cryogel preparation either via in situ covalent cross-linking that applies a number of chemical cross-linkers [7,8,9,10,11] or physical gel formation via hydrogen bonding and the chelation process [6,7]. A convenient way to perform cryogel synthesis includes radical polymerization of several monomers with different functionalities allowing for desirable physico-elastic and biochemical properties [9,11,12,13]. 

Hydrogels obtained from natural substrates, such as gelatine (Gel), heparin, chitosan, hybrid gelatin/heparin microspheres loaded with insulin-like growth factor-1 (IGF-1) [12,13,14] and dextran (Dex) [15] are among a group of materials with a wide application profile, especially in the medical and pharmaceutical area, due to their biocompatibility, nontoxicity, sufficient mechanical strength, stability etc. [14,15]. Compositions of Gel have been widely utilized with different polymers and cross-linking agents [12,16,17,18] for the synthesis of scaffolds for medical applications and tissue engineering [14,18,19,20]. There are a few pharmacopeia monographs for the specification of gelatin and matrixes from it [21].

Dextran is a microbiologically produced, renewable, commercially available polysaccharide [22] that can be used for the creation of a drug-delivery systems and materials for hydrogels in biomedicine due to its excellent hydrophilic nature and biocompatibility [23,24,25,26,27,28,29,30]. Dextran dialdehyde (DDA) due to its relatively high molecular weight, non-toxic nature, biocompatibility and biodegradability has a great potential as a cross-linker for different polymers [23,24,25,26], including the preparation of Gel microspheres [27,28,29,30].

Water has a great impact on the mechanical properties and stability of materials [31,32,33,34]. At the standardization stage of the material of biomedical application, the remaining humidity parameter significantly affects the storage conditions and shelf life of polymeric materials due to ageing-related changes that are both physical (the migration of filaments, rearrangements of amorphous regions and crystallization) and chemical (oxidation) [32]. The absorbing capacity of the material for water in the liquid and vapour phases is among the main requirements for wound dressing [35].

Interactions between solid-state materials and water vapour takes place during various stages of preparation and storage. Water vapour uptake may result in the decomposition and chemical and physical changes of the material [36,37]. The surface properties of the material can be investigated by the determination of sorption isotherms, which show the relationship between the water content of the material and the equilibrium humidity [38,39,40].

Many theoretical, partially theoretical and empirical models can be used for the mathematical explanation of moisture sorption isotherms. The Brunauer, Emmett and Teller (BET) model, which builds on the multi-layer adsorption of water vapour, is among the best; however, the application range of BET is limited to relative water vapour pressures up to 0.3–0.5. The GAB (Guggenheim, Anderson and de Boer) model can be used for the evaluation of sorption isotherms over a wider variety of relative humidity [41,42,43,44].

The main focus of this investigation was to study the water vapour sorption properties of new cryogels, prepared by green chemistry crosslinking of Gel by DDA. The equilibrium water sorption isotherms were determined by two independent methods: static-gravimetric and dynamic vapour sorption (DVS), and the obtained data were evaluated applying the BET and GAB models. Structural differences in the synthesized cryogels were investigated using Confocal Laser Scanning Microscopy. 

## 2. Results and Discussion

Cryogels based on DDA and Gel were synthesized and characterized according to the procedure described in our previous article, and a mechanism of interaction between macromolecules was proposed [45]. Schiff’s base group formation and comprehensive characterization of functional groups of the cryogels of DDA:Gel with mass ratios of 1:1, 1:2 and 1:3 were previously studied using FTIR and HNMR spectroscopy [45]. The elasticity modulus of these cryogels was in the range of 0.66 to 2.8 kPa [45]. Moreover, we estimated the biocompatibility of cryogels based on variations of chitosan, gelatine and dextran on fibroblasts NIH/3T3 cell line [9]. The proliferation of cells visualised by confocal microscope cell observer SD Carl Zeiss. It was observed that the increase in content of Gel led to the improvement in its biocompatibility [9]. Therefore, it was important to comprehensively characterise the surface properties of cryogels in order to determine the possible correlation between them and the biocompatibility of the material. The sorption isotherms were obtained for all samples: Gel, Dex, DDA (as components for cryogels preparation) and DDA:Gel cryogels with different ratios of DDA to Gel (1:1, 1:2 and 1:3), prepared according to the method described in Section 4.2. 

The initial water content of the cryogels was estimated from the drying step of each experiment and did not exceed 2.5 *w*/*w* percent. The majority of the experiments were performed by the static-gravimetric method; however, to confirm the reliability of the experimental data, some of the samples were also analysed using the DVS method. The experimental sorption isotherms for the DDA:Gel 1:3 cryogel determined by two independent methods are presented in Figure 1.

Figure 1 shows that the experimental sorption isotherms determined by two independent methods are almost identical. All samples studied show similar patterns during the sorption process. The experimental equilibrium water uptake data are summarized in Table 1 (±S.D. did not exceed 3%).

The data in Table 1 show that sorption capacities of the studied materials are rather similar, and the difference in water uptake is not significant. In general, at the same humidity, the moisture content of DDA:Gel gels is higher the more Gel is contained in the mixture (exception 33% RH). The action of water vapour upon the materials produced from pure Gel has been studied for more than one hundred years, and our experimental data are in good agreement with the published results [13,18,46,47,48]. The water uptake data obtained for pure and modified Dex gels are also in good agreement with the published data [15,49,50,51]. However, at high humidities (93% and 100%), the partial collapse of DDA-Gel gels was observed during incubation (the illustration is given in Figure 2). 

Similar phenomena were observed by other researchers [52,53] and explained by the pore network’s collapse at certain conditions due to different ratios of cross-linker to monomer or other factors. In our case, the high humidity treatment of dried cryogels provides the adsorption of water and, therefore, facilitates flexibility of weakly cross-linked polymer chains. This results in additional polymer–polymer interactions (electrostatic interactions between oppositely charged groups and the formation of Schiff’s groups as a product of reaction between available free aldehyde and amino groups). 

Thus, this leads to the formation of a more compact and possibly denser crosslinked 3D structure, which was indirectly confirmed by a 10% increase in the yield of the gel fraction compared to the initial high humidity untreated samples [45]. The isoelectric point (IEPs) of Gel-DDA (1:1), (2:1) and (3:1) shifted from 7.0 for pure Gel to 4.59, 6.11 and 4.045, respectively [9]. This is an indication that OxD had an excess of aldehyde groups and that the formation of Schiff’s base groups resulting in the additional cross-linking of polymers led to decrease of amino groups number. Therefore, a relative increase in carboxyl groups was observed, which was confirmed by an increase in the negative zeta potential at pH 7.2 [9].

Experimental sorption isotherms were analysed by BET and GAB models. BET theory was applied for the water activity range 0.1 ≤ *p*/*p_o_* ≤ 0.5, and the following equation was used:(1)$$\frac{1}{W\left[\left(\frac{p_{o}}{p}\right)-1\right]}=\frac{1}{W_{m}C}+\frac{\left(C-1\right)}{W_{m}C}\left(\frac{p}{p_{o}}\right),$$
where *W* (g/g) is the experimental equilibrium water content of the sample, *p* and *p_o_* are the equilibrium and the saturation pressure of water at the temperature of adsorption, respectively; *W_m_* is the monolayer capacity; and *C* is the BET constant, related to the water molecule’s binding energy. The *C* is a constant at any given temperature and is given by the equation:(2)$$C=exp(\frac{E_{1}-E_{\mathrm{EV}}}{RT})\;.$$

Here, *E*_1_ is the heat of adsorption for the first layer, and *E_EV_* is the heat of evaporation of pure water at a given temperature, *R* is the molar gas constant (8314 J mol^−1^K^−1^), and *T* is the temperature for sorption determinations (in our case, 298.15 K). The specific surface area (*S*) for the investigated samples was calculated according to the following equation:(3)$$S=\frac{W_{m}N_{a}A}{M},$$
where *W_m_* is the monolayer capacity, *N_a_* is Avogadro’s number, *A* is the cross-sectional area of adsorbate (for water 10.8 A^2^ [54]), and *M* is the molecular weight for water. The results for the BET calculations (the regression coefficients were greater than 0.994 for all samples) are given in Table 2.

As shown in Table 2, both the monolayer capacity (*W_m_*) and surface area (*S*) are dependent on the ratio between Gel and DDA in the composition. The surface area of freeze dried 4% dextran aerogel (311 m^2^/g) is almost the same as for DDA:Gel 1:1 (312 m^2^/g). Cryogels with higher gelatine content DDA:Gel 1:2 (236 m^2^/g) and DDA:Gel 1:3 (250 m^2^/g) show values almost identical to 4% gelatine aerogel (248 m^2^/g) prepared in a manner that is identical to that of cryogels. Our data are in good agreement with data obtained for gelatine based hydrogels for which the authors stated that the water adsorption decreased with an increase in the gelatine content [5]. 

The difference (*E*_1_ − *E_EV_*) is a parameter determining the energy of interaction between the surface of cryogels and the first monolayer of adsorbed water. Larger values of (*E*_1_ − *E_EV_*) reveal stronger forces of hydration between the cryogel surface and molecules of water. Two cryogels DDA:Gel 1:2 and DDA:Gel 1:3 show high interaction energy (from 15.8 to 19.89 kJ/mol). These values are high enough for the formation of hydrogen bonds between the sample and water molecules on the surface, and similar results were observed for other types of porous materials [55]. 

Three samples, Dex, DDA and DDA:Gel 1:1, showed negative values for (*E*_1_ − *E_EV_*), indicating an energetically weak water binding capacity [56]. Figure 3 shows microphotographs of cryogels DDA-Gel 1:1, 1:2 and 1:3. The comparison of CLSM microphotographs of hydrated cryogels indicates that the increase of gelatine content in DDA-Gel composition of samples led to an increase of the pore size. In the reaction mixture of DDA and Gel with a weight ratio 1:1, the probability of interaction between amino functionality and aldehyde is much higher due to a relative excess of aldehyde groups compared to the composition DDA-Gel 1:3. 

Therefore, presumably a very weak 3D network is formed just before the freezing of the system, and this, in turn, affects the nucleation mechanism of ice crystal formation. Additionally, the increase in the amount of gelatine in the composition results in an increase in the total quantity of functional groups, each of them having the multilayer of hydrated shells, which significantly influence the mechanism and rate of seeds crystal growth.

The porosity difference between the three compositions can also be related to the presence of a different amount of ionic groups in the gelatine structure, which, in turn, possesses a larger hydrated shell compared to non-ionic groups. The energy of interaction of water with charged polymers should be stronger than with noncharged polymers [57]. Such a structure difference can be a possible reason for the difference in *E*_1_ values. The formation of hydrogen bonds in pores of smaller size can lead to a higher density of this type of bonds per unit of material surface area.

As mentioned in the introduction, the GAB model is more beneficial in the regression analysis of water sorption experimental data compared with BET model, because it can be applied for the whole range of water activities. In contrast to BET where the sorption state of the sorbate molecules in the layers beyond the first is the same but different in the pure liquid state, the GAB model introduces a second sorption stage for water molecules, which leads to the appearance of the additional parameter *K*. This parameter measures the difference between the standard chemical potentials of the molecules in the second sorption stage and in the pure liquid [43]. By use of the “Origin 7.0 Data Analysis” software, a non-linear least squares fitting (NLSF) was performed to determine the parameters for the GAB model fitting equation:(4)$$W=\frac{W_{GAB}C_{GAB}K_{GAB}a_{w}}{\left(1-K_{GAB}a_{w}\right)\left(1-K_{GAB}a_{w}+C_{GAB}K_{GAB}a_{w}\right)}$$
where *W* is the moisture content of the material on a dry basis, *C_GAB_* is the Guggenheim constant related to heat of sorption, *a**_w_* is the water activity (equal to relative *p*/*p_o_* for BET model), *K_GAB_* is the constant related to multilayer molecular properties, and *W_GA_*_B_ is the moisture content of monolayer similar to *W_m_* in a BET theory [43]. 

As shown in Table 3, the use of the *GAB* model for the treatment of experimentally obtained sorption isotherms shows a perfect fit through the entire humidity range.

It is known that the parameter *C_GAB_* has a mostly enthalpic nature and is a measure of the strength of water binding to the primary binding sites. Large *C_GAB_* values indicate that water is strongly bound in the monolayer, and the difference in enthalpy between the monolayer molecules and the subsequent layers is larger. The results given in Table 3 show that cryogels with gelatine in the composition show higher levels of *C_GAB_* (from 29.8 to 33.0), which is in agreement with CLSM micrographs. Despite the fact that the regression coefficients were close to unit, the errors in the calculated *C_GAB_* values were very high (except for DDA), which is likely due to the possibility of chemical changes on the cryogels surface during water vapour sorption. 

Parameter K contains an important entropic part, while its enthalpic part is less compared with that in *C_GAB_*, due to the considerably lower interaction enthalpy of the multilayer molecules with the sorbent [58]. In the case of K = 1, the difference between multilayer molecules and bulk liquid molecules vanishes [42,43]. The calculated parameter K (Table 3) is rather closed to units, which indicates that properties of water molecules on a surface are close to those in bulk. These values were between 0.879 (Dextran) and 0.946 (DDA:Gel 1:1), which is in good agreement with the published data [39,40].

The thermogravimetric analysis (TGA) data for all types of cryogels are presented in Appendix A. TGA analysis of freeze dried Gel-DDA 3:1 cryogel illustrated about 8.8% of strongly bound water (Appendix A). The significant loss of weight for cryogel OxD-Gel 1-2 and GelGA under heating up to 200 °C might be related to the dehydration reaction of hemiaminal into the Schiff’s base as was discussed in our previous study [45], whereas OxD-Gel 1:3 has much fewer hemiaminal groups and is more stable in this temperature region. The cryogel samples OxD-Gel 1:3, OxD-Gel 1:2 and OxD-Gel 1:1 stored at RT contain 7.9%, 6.1% and 8.8% of unbound water, respectively, which was confirmed by mass decline under 100 °C. 

The decomposition temperature for OxD-Gel cryogels with the compositions 1:1, 1:2 and 1:3 were 253, 260 and 269 °C, respectively (Appendix A). The heat treatment of cryogel samples in the region from 125 to 500 °C lead to weight loss of 67%. The stability of the material and decomposition temperature of cryogel based on gelatin cross-linked by GA is comparable with OxD-Gel samples (Appendix A). One can conclude that developed system based on gelatin cross-linked by OxD is better as OxD is not toxic and is bio-compatible. Differential scanning calorimetry (DSC) was utilised to understand the physicochemical process taking place after high humidity treatment. 

For example, a glass-transition temperature (Tg), and decomposition temperature (Tdecom) for various combinations of Gel, with different concentrations of DDA or GA were evaluated. As one can observe, the increase of gelatin content in the cryogel resulted in increase of bound water, which increased the heat capacity to 268.6 J/g and 354 J/g for Gel-DDA 1:1 and Gel-DDA 3, respectively (Appendix A). The Gel DDA 1:1 100% RH contains bound water with an evaporation temperature of 71 °C and heat capacity of 268.6 J/g (51.92 °C) (Appendix A). 

Comparison of Gel:DDA 1:1 and Gel:DDA 3:1 indicated that weakly bound water began to evaporate at 45.7 and 37.6 °C indicating that gelatine’s polar groups are predominantly responsible for weakly bound water (Appendix A). Thus, the maximum of endothermic peaks for Gel:DDA 1:1 and Gel:DDA 3:1 were at 110 and 94.7 °C, respectively. To confirm that these endothermic peaks related to the water and not to the degradation of the material, several cycles of cooling/heating were performed (Appendix A). The material Gel:DDA 2:1 and other samples were stable upon heating up to 200 °C, and any phase transitions was observed for dehydrated samples of cryogel (Appendix A).

## 3. Conclusions

The water sorption properties of macroporous cryogels, based on gelatine and dextran dialdehyde were determined experimentally. The BET and GAB models were successfully applied to the treatment of experimental data and estimation of important physicochemical parameters. This study has shown the dependence of surface properties upon the composition of cryogels; a higher content of Gel in the material can lead to the possible formation of H-bonds, which is an interesting observation for further use of these new materials for tissue engineering. 

One of the main roles of artificial tissues is protection of the body against dehydration. In order to fulfil this function, the gel should be strongly hydrated. According to TGA the decomposition temperature for OxD-Gel cryogels with ratios of 1:1, 1:2 and 1:3 were 253, 260 and 269 °C, respectively. Cyclic DSC analysis confirmed that the endothermic peaks at about 100 °C related to water evaporation and not to the degradation of material. 

The material Gel:DDA 2:1 and other samples were stable upon heating up to 200 °C, and phase transitions of chemically cross-linked cryogels after the first heating /cooling cycle (dehydrated samples) were observed. From the obtained data, one can propose optimal storage conditions of porous material. Our future studies will be devoted to the testing of these cryogels loaded with growth factors in vitro and then in vivo for regenerative medicine and the estimation of a relation between the structure effect and a cell’s proliferation efficiency. 

## 4. Materials and Methods

### 4.1. Chemicals

The salts: LiCl, HCOOK, MgCl_2_, K_2_CO_3_, NaBr, NaCl, (NH_4_)_2_SO_4_, KCl and KNO_3_, which were used for the preparation of saturated solutions to create a certain level of humidity, were purchased from Sigma. Dextran T10, T40 and T2000 (Mw 10, 40 and 2000 kDa) were obtained from Pharmacia (Uppsala, Sweden), gelatine 45% *w/v* solution (from cold water fish) and glutaraldehyde (GA) 25% *v/v* aqueous solution were purchased from Sigma (Steinheim, Germany). The DDA was prepared and characterized according to previously published methods [45,59]. 

### 4.2. Preparation of Cryogels

The following cryogels were prepared from Gel, Dex, DDA and their mixtures. The preparation of the control samples (aerogel-like structure) from Gel, Dex T40 or DDA was performed with the following procedure: the glass tubes (diameter of 7 mm) were filled with 0.5 mL of 4 wt % polymer mixture, and the samples were rapidly cooled and frozen at −12 °C in cryostat, and then the specimen were kept in a freezer at −22 °C for 24 h. The frozen cryogels were lyophilized to constant weight. 

For the preparation of cryogels from mixtures, preliminary cooled Gel and DDA solutions were properly mixed at given ratios DDA:Gel (% *w*/*v*): 2:2, 1.33:2.66, 1:3, respectively, and transferred into glass tubes (diameter 0.7 cm). These tubes were consecutively frozen in an ethanol cryobath at −12 °C and then in an air cryobath at −12 °C overnight. Finally, the frozen cryogels were placed into the freezer and then lyophilized to constant weight for approximately 24–36 h. The dried cryogels were kept at room temperature in a desiccator. The detailed procedure and properties of cryogels are described in our previous publication [45].

### 4.3. Confocal Laser Scanning Microscopy (CLSM) Images of Cryogels

CLSM can be used for characterization of cryogel structure during swelling under different humidities at the atmospheric pressure. Thin slices of lyophilized samples were cut by blade and were fixed in 5% *v/v* GA in PBS buffer overnight. Cryogels were incubated in distilled water to remove unreacted GA and stained with Rhodamine B solution overnight [7]. Unbound Rhodamine B was washed out with water. Confocal laser scan microscopy (CLSM) images were obtained using a Leica TCS SP5 using objective lens ×10, ×20, ×40 and ×63. The 488 and 530 nm excitation and emission wavelengths were applied. Microphotographs were produced by optical sectioning in the xy-planes along the *z*-axis with 30–70 optical sections with 1 μm intervals.

### 4.4. Water Vapour Sorption Methods

Two methods were used for determinations of water sorption isotherms: static-gravimetric and the dynamic vapour sorption (DVS). The investigated samples, cylindrically shaped with weight 0.01–0.03 g, were dried in a desiccator with drying agent (silica gel blue) until constant weight prior to water sorption determinations. High precision mechanical balance (Mettler M5 S/A, Greifensee, Switzerland) with an accuracy of ±2 µg and the precision of ±1 µg was used for weight control. The samples in open weighing vials were placed in an air-sealed glass chamber, where the constant relative humidity (RH) was achieved by mounting hygrostats filled with either distilled water (100% RH) or saturated salt solutions [60]. 

The samples were kept in humidity chambers until equilibrium (constant weight), which took 3–7 days, depending on the sample nature and RH. All experiments were performed in a thermostated room at 293° K. At least three parallel determinations were performed for each sample at each humidity. The details of DVS method are given in a previous publication [38]. The DVS was used as a control, and the determinations were performed on the DVS Advantage 1 instrument (Surface Measurement Systems, London, UK) at 293° K.

### 4.5. Thermogravimetric Analysis

The degradation of OxD-Gel cryogels were determined using TGA (Q500 analyser from TA Instruments). The operation was performed from 40 to 500 °C at a heating rate of 10 °C/min. 

### 4.6. Differential Scanning Calorimetry

The composite and control samples (cryogel without particles) were analysed under nitrogen during heating from −80 to 170 °C at a heat flow 10 °C/min. The cyclic method was applied for all samples to estimate irreversible changes of the structure under heating and loss of water.

## Figures and Tables

**Figure 1 gels-07-00159-f001:**
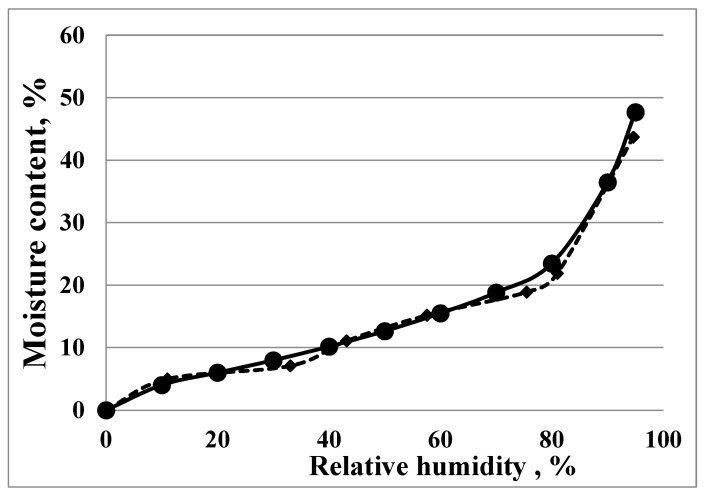
Experimental sorption isotherms, obtained by two independent methods: (-●-) DVS and (-♦-) static-gravimetric, for DDA:Gel 1:3 cryogel.

**Figure 2 gels-07-00159-f002:**
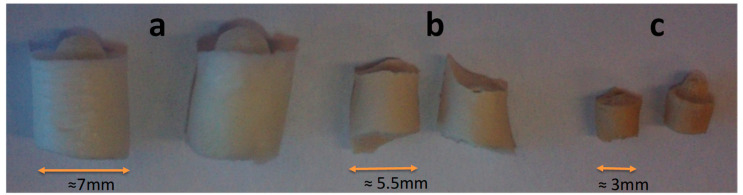
Photo of DDA-Gel 1:3 samples after incubation at high humidity and following drying in desiccator at room temperature: (**a**) initial gel, (**b**) after 93% RH, (**c**) after 100% RH.

**Figure 3 gels-07-00159-f003:**
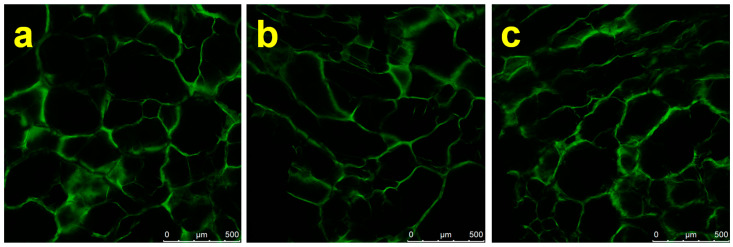
CLSM micrographs of the macroporous cryogels DDA-Gel: (**a**) 1:1, (**b**) 1:2, and (**c**) 1:3.

**Table 1 gels-07-00159-t001:** Equilibrium moisture content (g/100 g) of cryogels at different relative humidities (RH). ±S.D. did not exceed 3%.

RH, %	Gel	Dex	DDA	DDA:Gel 1:1	DDA:Gel 1:2	DDA:Gel 1:3
11	4.4	3.1	2.3	2.7	4.4	5.0
33	9.0	8.6	7.1	8.0	7.9	7.1
43	10.6	11.1	10.3	10.2	10.5	11.1
59	12.3	14.8	14.8	14.4	14.0	15.2
75	18.4	21.4	19.5	16.8	16.8	18.9
81	23.3	24.6	25.1	20.4	20.4	21.9
95	49.1	39.8	43.4	40.7	41.7	43.7
100	117.7	60.2	83.0	93.0	106.7	111.8

**Table 2 gels-07-00159-t002:** Surface characteristics for the investigated samples (BET); ±S.D. did not exceed 3%.

Sample	*W_m_*, g/g	*S*, m^2^/g	*C*	*E*_1_, kJ/mol	*E*_1_ − *E_EV_*, kJ/mol
Gelatine	0.069	248	11.17	59.80	15.80
Dextran	0.086	311	3.83	33.28	−10.72
DDA	0.092	334	2.24	20.00	−24.00
DDA:Gel 1:1	0.086	312	3.78	32.95	−11.05
DDA:Gel 1:2	0.065	236	11.38	60.26	16.26
DDA:Gel 1:3	0.069	250	13.18	63.89	19.89

**Table 3 gels-07-00159-t003:** GAB fitting parameters for the experimental sorption isotherms.

Sample	*W_GAB_*, g/g	*K*	*C_GAB_*	Regression Coefficient
Gelatine	0.0591 ± 0.0027	0.926 ± 0.006	29.8 ± 23.7	0.998
Dextran	0.0721 ± 0.0061	0.879 ± 0.011	11.51 ± 9.17	0.994
DDA	0.0614 ± 0.003	0.923 ± 0.003	9.02 ± 0.86	0.998
DDA:Gel 1:1	0.0506 ± 0.003	0.9456 ± 0.003	32.2 ± 67.2	0.996
DDA:Gel 1:2	0.0583 ± 0.004	0.9053 ± 0.012	33.0 ± 42.4	0.993
DDA:Gel 1:3	0.0646 ± 0.005	0.897 ± 0.013	23.4 ± 24.0	0.993

## Data Availability

Supporting reported results can be found.

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
