# Peer review of "Water Uptake as a Crucial Factor on the Properties of Cryogels of Gelatine Cross-Linked by Dextran Dialdehyde"

_gels, 2021, doi:10.3390/gels7040159_

Round 1

Reviewer 1 Report

In this manuscript, the authors water vapor sorption properties of new cryogels, prepared by green chemistry crosslinking of gelatine by dextran dialdehyde. The methods are well described in details and results are interesting and scientifically sound. However, there are still some questions and unexplained results that needed to be solved. So the reviewer suggests to accumulate more data to complete the manuscript with some revision.

  1. The reviewer recommends to conduct swelling ratio measurement of cryogels to show more information about water sorption properties.
  2. More information about gelatin and dextran dialdehyde are needed in the introduction section. It is difficult to understand why the author chose it exactly.
  3. The reviewer suggests to measure the pore size or porosity of new cryogels (DDA-Gel: 1:1; 1:2; 1:3) by SEM or other methods that author prefers since the pore size or porosity of new cryogels can affect the water sorption properties.
  4. The CLSM images in figure 3 are not clear enough to observe. Especially, figure 3b needs to be changed to better image.

Author Response

  1. We thank the reviewer for valuable feedback that aimed to improve quality of the paper.
  2. Comment. The reviewer recommends to conduct swelling ratio measurement of cryogels to show more information about water sorption properties.

We have published swelling data in water and in phosphate buffer in a previous paper (Berillo and Volkova 2014), and this ref was in the manuscript. As the swelling of the cryogel  in liquid water  and vapor water adsorption are different processes, we did not include this data in the current manuscript.

Comment. More information about gelatin and dextran dialdehyde are needed in the introduction section. It is difficult to understand why the author chose it exactly.

We think that these new cryogels, prepared by green chemistry crosslinking, that degradation rate  can change vs pH and that can be of interest for many areas of the research. The introduction part included almost 40 references. Additional ref were introduced. 

Milakin, K. A., Acharya, U., Trchová, M., ZasoÅ„ska, B. A., & Stejskal, J. (2020). Polypyrrole/gelatin cryogel as a precursor for a macroporous conducting polymer. Reactive and Functional Polymers, 157, 104751.   Abudula, T., Colombani, T., Alade, T., Bencherif, S. A., & Memić, A. (2021). Injectable Lignin-co-Gelatin Cryogels with Antioxidant and Antibacterial Properties for Biomedical Applications. Biomacromolecules. Huang, Y., Zhao, X., Zhang, Z., Liang, Y., Yin, Z., Chen, B., ... & Guo, B. (2020). Degradable gelatin-based IPN cryogel hemostat for rapidly stopping deep noncompressible hemorrhage and simultaneously improving wound healing. Chemistry of Materials, 32(15), 6595-6610.

Comment .The reviewer suggests to measure the pore size or porosity of new cryogels (DDA-Gel: 1:1; 1:2; 1:3) by SEM or other methods that author prefers since the pore size or porosity of new cryogels can affect the water sorption properties.

We are agree with a reviewer, that the porosity will affect the water sorption properties, but, unfortunately, the manuscript can’t involve all possible methods of investigation of present systems.

Comment. The CLSM images in figure 3 are not clear enough to observe. Especially, figure 3b needs to be changed to better image.

We are agree with a reviewer, and it was changed with a better quality.

Reviewer 2 Report

General remarks

To my astonishment, it turned out that I have already reviewed for ‘GELS’ in 2019 virtually the same manuscript, and my recommendation about 3 years ago was as follows: “the manuscript should be subjected to certain enhancements (minor revision) before to be accepted for publication”. However, the paper has not been published (the reasons are unknown for me, as well as I did not receive the authors’ response for the reviewers’ remarks).

Now such recommendation is, in fact, very close, i.e. the paper (gels-138826) can be of interest for the special issue related to cryogels, but can be accepted for publication after minor revision. The revisions are required, since the authors did not took into an account some previous my remarks (see below), as well as some terminology of the paper should be polished.

Particular remarks

Tables 1 and 2:

The experimental error (+/-) must be indicated in the Table 1; otherwise the data look as the result of a single experiment in every point without any reproducibility. The same remark is also related to the experimental data given in Table 2 (this remark was also done 3 years ago, but, unfortunately, it turned out to be without the authors’ response).

Page 5:

Additionally, the increase of amount of gelatin in the composition results in increase of total quantity of functional groups, each of which has multilayer of hydrated shell, which significantly influence the mechanism of NANOCRYSTAL formation  –  I did not find in the manuscript any information about some kind of NANOCRYSTAL; dear authors, where you detected the presence of nanocrystals in your systems? (this remark was also done 3 years ago, but, unfortunately, it turned out to be without the authors’ response)

Page 6:

Despite of the fact that regression coefficients were close to unit, the error in calculated CGAB values were very high (except DDA), probably due to the possibility of chemical changes of cryogels surface during water vapour sorption  –  what does it mean “chemical changes of cryogels surface” (???); it is unclear why the authors think that upon the water interaction with gelatine at room temperature any sort of chemical reaction can occur; if such reaction or reactions are able to proceed, what chemical experimental data testify about it? (this remark was also done 3 years ago, but, unfortunately, it turned out to be without the authors’ response)

Terminology:

 (i) Title of the paper

“Water Uptake as a Crucial Factor on Properties of Cryogels of Gelatine Linked by Dextran Dialdehyde”  

From the chemical point of view it will be better to write ‘Cross-Linked’ instead of simply “Linked”.

(ii) Through the text the authors use such term as “dry cryogel”. The tem is insufficiently correct, since any gel (including cryogels, as well) is the swollen polymeric network. It means that none gel (of course, including cryogels) can be dry. Therefore, the respective samples can be ‘dried’ ones rather than ‘dry (cryo)gels’.

Author Response

We thank the reviewer for the time devoted for a revision and for the comments.

Comment. To my astonishment, it turned out that I have already reviewed for ‘GELS’ in 2019 virtually the same manuscript, and my recommendation about 3 years ago was as follows: “the manuscript should be subjected to certain enhancements (minor revision) before to be accepted for publication”. However, the paper has not been published (the reasons are unknown for me, as well as I did not receive the authors’ response for the reviewers’ remarks).

We are really sorry, that the reviewer feels in this way. We were trying to do our best to improve the quality of our manuscript, according to the comment, which we got 2019, but probably our reply to comments were not transferred by the editor. This manuscript was additionally supplemented with TGA and DSC data.

Particular remarks

Comment. Tables 1 and 2: The experimental error (+/-) must be indicated in the Table 1; otherwise the data look as the result of a single experiment in every point without any reproducibility. The same remark is also related to the experimental data given in Table 2

We did the experiment in triplicate and used average data in tables. The “±S.D. did not exceed 3%)” in Table 1 and the “regression coefficient were greater than 0.994 for all samples” are enough for our purposes.

Comment Page 5: “Additionally, the increase of amount of gelatin in the composition results in increase of total quantity of functional groups, each of which has multilayer of hydrated shell, which significantly influence the mechanism of NANOCRYSTAL formation”  –  I did not find in the manuscript any information about some kind of NANOCRYSTAL; dear authors, where you detected the presence of nanocrystals in your systems? (this remark was also done 3 years ago, but, unfortunately, it turned out to be without the authors’ response)

We think, that this can be the reasonable explanations of our results, what can have happened in such a system at this conditions. Nevertheless, we agree that it was not supported properly with experimental data, therefore this paragraph was paraphrased in a better way now.

comment Page 6:“Despite of the fact that regression coefficients were close to unit, the error in calculated CGAB values were very high (except DDA), probably due to the possibility of chemical changes of cryogels surface during water vapour sorption”  –  what does it mean “chemical changes of cryogels surface” (???); it is unclear why the authors think that upon the water interaction with gelatine at room temperature any sort of chemical reaction can occur; if such reaction or reactions are able to proceed, what chemical experimental data testify about it? (this remark was also done 3 years ago, but, unfortunately, it turned out to be without the authors’ response)

Last time I did not have access to e-mail for a long time and therefore I could not help and  the coauthour could not properly address the question. We have published previously that even a moderate heating at 60C leads to additional cross-linking of the dry cryogel that was supported by shift of some bonds in FTIR spectra, and control experiments ascribed. The chemical changes upon heat treatment was also confirmed via following swelling experiment and comparison of swelling degree with untreated cryogel (Berillo and Volkova 2014). What happens upon water adsorption / desorption,  unreacted aldehyde groups can significantly faster be oxidised to carboxyl groups compare to dry state. This in turn can lead to electrostation interations between oppositely charged functional groups, which in tern effect some conformation of a relatively weakly crosslinked polymer network. Moreover, dextran dialdehyde forms hemiacetal groups, that was confirmed by 1H-NMR (Berillo and Volkova 2014). These groups can be transformed to aldehyde group in presence of water, specially in presence of CO2 the medium is slightly acidic.  Aldehyde groups can react with free amino groups, providing additional cross-linking. In order to confirm this with experimental data microcalorimetric analysis should be performed, but at present we don't have access to high precision instrument.

Terminology: (i) Title of the paper

“Water Uptake as a Crucial Factor on Properties of Cryogels of Gelatine Linked by Dextran Dialdehyde”  

From the chemical point of view it will be better to write ‘Cross-Linked’ instead of simply “Linked”. We agree with the comment and thankful to the reviewer.

(ii) Through the text the authors use such term as “dry cryogel”. The tem is insufficiently correct, since any gel (including cryogels, as well) is the swollen polymeric network. It means that none gel (of course, including cryogels) can be dry. Therefore, the respective samples can be ‘dried’ ones rather than ‘dry (cryo)gels’. We agree with the point and corresponding changes in the text was done.